# Influence of Some Spaghetti Processing Variables on Technological Attributes and the In Vitro Digestion of Starch

**DOI:** 10.3390/foods11223650

**Published:** 2022-11-15

**Authors:** Mike Sissons, Silvia Cutillo, Narelle Egan, Asgar Farahnaky, Agata Gadaleta

**Affiliations:** 1NSW Department of Primary Industries, Tamworth Agricultural Institute, 4 Marsden Park Road, Calala, NSW 2340, Australia; 2Department of Soil, Plant and Food Sciences (DiSSPA), University of Bari Aldo Moro, Via G. Amendola 165/A, 70126 Bari, Italy; 3Biosciences and Food Technology, School of Science, RMIT University, Bundoora West Campus, Melbourne, VIC 3083, Australia

**Keywords:** starch digestion, durum wheat, pasta quality, extrusion, pasta cooking

## Abstract

Durum semolina spaghetti is known to have a low-moderate glycaemic index but the impact of various processing variables during the manufacture and cooking of pasta does affect pasta structure and potentially could alter starch digestion. In this study, several process variables were investigated to see if they can impact the in vitro starch digestion in spaghetti while also monitoring the pasta’s technological quality. Cooking time had a large impact on pasta starch digestion and reducing cooking from fully cooked to al dente and using pasta of very high protein content (17%), reduced starch digestion extent. The semolina particle size distribution used to prepare pasta impacted pasta quality and starch digestion to a small extent indicating a finer semolina particle size (<180 µm) may promote a more compact structure and help to reduce starch digestion. The addition of a structural enzyme, Transglutaminase in the pasta formulae improved overcooking tolerance in low protein pasta comparable to high protein pasta with no other significant effects and had no effect on starch digestion over a wide protein range (8.6–17%). While cold storage of cooked pasta was expected to increase retrograded starch, the increase in resistant starch was minor (37%) with no consequent improvement in the extent of starch digestion. Varying three extrusion parameters (die temperature, die pressure, extrusion speed) impacted pasta technological quality but not the extent of starch digestion. Results suggest the potential to subtly manipulate the starch digestion of pasta through some processing procedures.

## 1. Introduction

Durum wheat (*Triticum turgidum* subsp. *durum*) is commonly used throughout the world to make pasta products. Pasta is rich (65–75%) in carbohydrates [1] and is known for its low-moderate glycaemic index (GI) [2]. Foods that reduce postprandial glycemic and insulinemic responses assist diabetics with glucose homeostasis [3]. Pasta products enriched with various ingredients (legumes, wholegrains, vegetables, resistant starch, etc.) have been created (for reviews see [4,5]) and some of these have reduced the low-moderate GI of pasta further but there is still a large variation in GI between pasta and the mechanism that underpins these differences are not well understood [2]. While the measurement of food GI is the best way to determine the glycaemic response to ingestion of a specific food, this method is expensive and resource-demanding and so in vitro methods have been developed to rank foods by predicting GI that correlates with the in vivo method [6] while also providing data on the kinetics of the starch digestion.

The manufacture of extruded spaghetti products involves first milling the grain into coarse semolina of particle size range typically 630 µm to <125 µm [7]. The semolina is then hydrated and mixed to obtain a dough with about 28–32% water content (14% mb). This transforms semolina into a homogeneous dough consisting of pea-sized lumps (of 2–3 cm diameter). The dough is then extruded under vacuum (5–10 MPa) at ~40–50 °C (single screw extruder) where the dough is developed in the extrusion barrel subject to shearing stress and then forced through the narrow aperture of a die to create the desired shape. Variation in extrusion parameters can impact pasta structure and technological properties [8]. The fresh pasta is then dried under carefully controlled temperature (40–85 °C) and relative humidity (70–90%) to produce a low moisture (~12%), dried product and this process results in the starch granules being embedded in a protein matrix [9,10]. For consumption, the dried product is cooked in boiling water for 8–14 min depending largely on the spaghetti diameter, protein content, etc. until al dente or to the fully cooked time (where the starch central core disappears). The pasta is strained and sometimes rinsed with water then consumed usually with other foods. During cooking starch gelatinisation is complete and protein coagulation occurs within the pasta structure with water acting as a plasticizer to ensure polymer mobility [9,11]. The structure of cooked pasta is a compact matrix of starch granules embedded in a protein network [12]. Modification of the pasta processing and cooking/storage conditions could alter the pasta structure and consequently influence starch digestibility and technological properties [9] as well as processing aids like transglutaminse [4] and these may be a way to further modulate pasta GI.

The purpose of this study was [1] to evaluate the impact of semolina particle size, extrusion conditions and transglutaminase addition on pasta technological properties (texture, cooking quality) and the pasta in vitro starch digestion and [2] the impact of cooking time and storage after cooking on pasta starch digestion to see if starch digestibility can be reduced.

## 2. Materials and Methods

### 2.1. Materials

Microbial transglutaminase (glutaminyl-peptide-amine γ-glutamyltransferase, E.C. 2.3.2.13) was a gift from IMCD Australia (Veron TG, Mulgrave, Australia). Semolina was obtained either from Durum Breeding Australia (variety DBA Bindaroi) or from a commercial source (Bellata Gold, Westdale, NSW, Australia). Gluten was isolated from a commercial semolina sample, freeze-dried and ground using a coffee grinder and sieved (500 μm mesh) as described previously [13]. Gluten was used to increase the protein content of semolina to desired levels. Commercial spaghetti (Barilla spaghetti n.5) was obtained from a local supermarket.

### 2.2. Analytical Methods

Flour water absorption was determined using a MicroDoughLAB (Perten Instruments, Macquarie Park, Australia) fitted with a 50 g bowl and mixing at 120 rpm to a target peak of 650 FU, in duplicate expressed on a 14% mb. Flour water absorption (FWA) at a target midline peak consistency of 665 Farinograph Units (FU), time in minutes to reach midline peak dough development (DDT), and FU loss in midline peak height to 5 min past peak (softening after 5 min) were recorded. Softening is regarded as an indicator of tolerance to mixing, with strong, tolerant doughs having low softening values. Semolina protein was determined by Dumas combustion using a Leco TruMax CN combustion nitrogen analyser (Leco Corp. St. Joseph, MI, USA) calibrated with sulfamethazine (N × 5.7). Semolina moisture was determined by AACC Method 44-15A [14]. Total starch and resistant starch (RS) content of ground spaghetti (coffee grinder, sieved across a 250 µm screen) were assayed in triplicate using Megazyme Total Starch and Resistant Starch kits; analysis procedure (a) was used for total starch (Deltagen Australia, Melbourne, Australia).

### 2.3. Spaghetti Preparation and Assessment

Single-screw extruder: Spaghetti was prepared based on Sissons et al. (2014) [15] but with adjustment for the amount of water added to make the dough based on water absorption differences in the samples caused by compositional differences corrected to a reference 29% water addition (290 mL water added to 1 kg purified semolina). In brief, semolina was mixed with water in a premixing chamber for 15 min then the crumbly dough transferred to the vacuum and temperature controlled (45 °C) extruder, fitted with a 1.82 mm teflon coated spaghetti die (Appar Laboratorio, Rome, Italy). After extrusion, the fresh spaghetti was transferred to a drying cabinet (TEC 2604, Thermoline Scientific Equipment, Smithfield, Australia) and held at 25 °C, 85% RH until all the samples were collected and drying commenced by increasing the temperature to 65 °C at 70% RH for 45 min then a period of 13 h at 50 °C, 80–70% rh followed by cooling to 25 °C, 55% RH for 4 h. Dried spaghetti (uncooked diameter 1.9 mm) was stored in sealed plastic bags at room temperature until required for analysis. For details on the twin-screw extruder used, see Section 2.5.4. All spaghetti samples were cooked to fully cooked time (FCT), the time taken for the central starch core to disappear using AACCI Approved Method 66-51.01 [14]. Cooking quality and texture and colour were assessed as reported earlier [16] except firmness was measured using AACC International Approved Method 66-52.01 [14] with a texture analyzer, TAXT2i (Stable Micro System, Godalming, UK). Firmness was as expressed as the peak force (F-PH) and work to cut (F-Area), measured on four separate cooks of each sample and the average value presented. Overcooking tolerance was measured by determining the firmness peak force after 10 min of overcooking (FCT + 10 min) in quadruplicate, calculated as [(Firmness at FCT − Firmness at FCT + 10 min)/Firmness at FCT) × 100]. Spaghetti stickiness was measured on five, 7 cm spaghetti strands cooked to their FCT, cooled in water for 2 min, drained and placed on the texture analyser platform fitted with the stickiness probe (HDP/PFS https://www.stablemicrosystems.com/adhesion-testing.html (accessed on 18 October 2022)). A retaining plate is placed over the spaghetti strands to stop them lifting when the probe retracts. The test begins after 3 min with the probe descending onto the strands at 5.0 mm/s and after contact, the probe is held for 2.0 s then retracted (1.0 mm/s) until the spaghetti is released. A 5 kg load cell using a force of 500 g was used. Stickiness was assessed as peak force (S-PH) and area under the force vs. time curve (S-Area) with a mean of three measurements presented. Spaghetti water absorption (WABS) was calculated as the change in spaghetti weight after cooking to FCT, expressed as a percentage of the uncooked weight and performed in duplicate. Cooking loss (CL) was determined on 5.0 g of spaghetti (3.5 cm long) cooked to FCT, drained, and rinsed with distilled water. The cooking water and rinsate were collected in pre-weighed beakers and evaporated to dryness in an air oven at 100 °C. Cooking loss was calculated as the change in weight after drying and expressed as a percentage of uncooked spaghetti weight. Measurements were performed in duplicate. The colour of uncooked (dry) spaghetti was analyzed for color space using a colourimeter CR-410 (Konica Minolta Sensing Inc., Osaka, Japan) calibrated with a white tile as a reference. Spaghetti strands, 7 cm sufficient to cover a Petri dish (10 cm × 1 0 cm plastic Petri dish on a white tile) one layer thick. Color parameters were L* (lightness) ranging from 0 (black) to 100 (white); a* (positive value is redness and negative value is greenness), and b* (positive value, yellowness; negative value, blueness). Data for color are the mean of three replicate readings taken along different parts of a spaghetti strand.

### 2.4. In vitro Starch Digestion of Spaghetti

Starch digestion of the samples was determined based on previous work [10]. 8 g spaghetti was cooked in 250 mL water to FCT, drained and cooled in water. Six spaghetti strands were trimmed to ~5 mm length. To standardize digestions an amount of cooked spaghetti calculated to contain 90 mg of starch for each sample (weight of uncooked spaghetti containing 90 mg starch) × (weight of cooked spaghetti/weight of uncooked spaghetti)) was subject to digestion. About 9–12.5 mm pieces were added to two 100 mL conical flasks (sample and one control-no enzymes added) to which 6 mL of pre-heated (37 °C) RO water was added and 5 mL of pepsin (Sigma P-6887 from gastric porcine mucosa) solution (1 mg/mL in 0.02 M HCl) except for control with 0.02 N HCL added. Flasks were incubated with shaking at 140 rpm for 30 min in a water bath held at 37 °C. To terminate the reaction, 5 mL of 0.2 M sodium acetate buffer (pH 6.0) was added to each flask followed by the addition of 5 mL α-amylase/amyloglucosidase (prepared fresh, 135 Units α-amylase, Sigma A6255 from porcine pancereas and 210 Units amyloglucosidase, Megazyme E-AMGDF) solution and 5 mL of buffer to controls then incubated for 360 min at 37 °C, 140 rpm. During the incubation, at intervals, a 0.1 mL aliquot was removed from the reaction mixture and mixed with 0.9 mL of ethanol (to terminate the enzyme reaction). This mixture was assayed for glucose using the Megazyme GOPOD reagent kit as per instructions. Absorbance at 510 nm was recorded using a UV mini-1240 Spectrophotometer (Shimadzu, Rydalmere, Australia). Glucose content (mg/mL) = Corrected sample absorbance [test sample absorbance-control absorbance]/Absorbance glucose standard.

The starch digested (%) was calculated as:Glucose content × 10 × 21 × (162/180) × (100/90)(1)
where 10 is the dilution factor (0.1 mL of reaction mix added to 0.9 mL ethanol); 21 dilution factor (1 mL to 21 mL reaction mix); 162/180 molecular weight ratio when converting from starch to glucose; 90 quantity of starch present in reaction mix in mg; 100 to convert to %. Replication was achieved by repeating the digestion of the same sample on a different test day with up to 2–4 replications per sample. The average of replicate values are presented. The incremental area under the digestion curves (AUC) was calculated.

Starch digestion data were fitted to a first-order equation:*Ct* = *C*∞ (1 − e−*kt*)(2)
where *Ct* is the percentage of starch digested at a given time (*t*), *C*∞ is the estimated percentage of starch digested at the end point of the reaction, and *k* is the starch digestion rate coefficient.

### 2.5. Experimental Trials

#### 2.5.1. Effect of Transglutaminase (TG) Addition in Pasta Making on Technological Quality and Spaghetti Starch Digestion

Commercial semolina control and two semolina protein extremes (low protein, 9.53%, 14% mb; high protein 16.88%, 14% mb) were used to assess the impact of TG addition at different protein levels. Previous work showed that the optimum amount of TG to add to semolina to increase dough strength was 0.5% *w*/*w* [17]. Each of the three semolina samples had TG added or not (control) during the preparation of spaghetti. The FWA was obtained and used to adjust water addition in the pasta making as higher protein increased FWA. Selected pasta quality evaluation methods were used to assess the impact of TG as well as the in vitro starch digestion (AUC) determined in duplicate.

#### 2.5.2. Effect of Spaghetti Cooking Time on Starch Digestion

The pasta was prepared from semolina with different protein contents (8.60%, 11.60%. 17.0%, 14% mb) from samples described previously [18]. The pasta was undercooked (FCT less 5 min) and overcooked (FCT plus 10 min) to represent typical cooking patterns of consumers and starch digestion was determined in duplicate. RS was measured on the ground, freeze-dried pasta.

#### 2.5.3. Effect of Spaghetti Storage Conditions after Cooking on Starch Digestion

A commercial spaghetti sample was cooked to FCT and stored either at 4 °C or 23 °C for 0, 48, and 72 h in humidified sealed plastic containers. Moisture loss in storage was minimal (0.94–1.18% after 24 h and 2.88–3.06% after 72 h) under these conditions. The RS was measured in uncooked pasta and cooked pasta stored at 0, 48 and 72 h. Starch digestion was performed on the stored pasta samples in duplicate.

#### 2.5.4. Effect of Spaghetti Extrusion Conditions on Starch Digestion and Technological Quality

Spaghetti was prepared from commercial semolina using a 16 mm barrel twin-screw extruder (Twin Screw Extruder; Thermo Fisher Scientific, Karlsruhe, Germany) to produce samples with the possibility to control multiple parameters such as:(1)Feed rate: the dry mix was fed into the twin screw feeder at various rates 40, 44, 36, 32 g/min.(2)Screw speed: The extruder screw speed was varied 150, 200, 250 and 300 rpm (using a feed rate of 40 g/min).(3)Die temperature: 61, 70, 75 and 85 °C (using feed rate of 40g/min and screw speed 200 rpm).(4)Die pressure: 14, 22, 28 and 32 bar (using feed rate of 40 g/min, die temperature 54–60 °C and screw speed 200 rpm).

Appendix A provides details of the sample treatment conditions. During extrusion, the required water was added using a peristaltic pump at the rate of 15 mL/min. Standard conditions were extruder die temperature 43 ± 3 °C using a 2 mm round die; feed rate of 40 g/min; screw speed 200 rpm. The extruded samples were collected and sealed in plastic bags. Fresh spaghetti was kept cool (24 h) before drying as described in Section 2.3.

#### 2.5.5. Effect of Semolina Particle Size on Spaghetti Technological Quality and Starch Digestion

Commercial semolina was sieved using a vibratory sieve shaker (Fritch, Analysette 3 Spartan, Chatswood, Australia) adjusting the amplitude to 2.0 mm with a run time of 3 min using stacked screens from 500, 425, 315, 250, and 180 μm sieve apertures. The amount of semolina retained on each screen was collected until about 2 kg of each fraction was obtained (except material on the 500 μm sieve was not used due to low yield and very high bran content). Material passing through the 180 μm screen was collected in the pan (<180 μm) is akin to flour. So for example, material retained on a 315 μm screen will have a particle size range from 315–425 μm and so on These samples are referred to as Control (unfractionated semolina), <180 μm, 180 μm, 250 μm, 315 μm and 425 μm and typical recovery for each of these fractions were 6.8, 10.5, 21.8, 38.8 and 17.4% with 4.7% retained on the 500 μm sieve. Each of these samples was evaluated for their farinograph water absorption and dough characteristics and this data was used to adjust water addition during pasta manufacture (1 kg scale). The pasta was assessed for technological quality and starch digestion.

### 2.6. Statistical Analyses

Data were analyzed using the statistical software GenStat version 19.1.0.21390 with a generalized linear model with a balanced design and the predicted means were tested for significant differences by the Least Significant Difference statistic (LSD), *p* < 0.05.

## 3. Results and Discussion

### 3.1. Effect of Semolina Particle Size on Spaghetti Technological Quality and Starch Digestion

It is known that the size and shape of the pasta subject to starch digestion (eaten) has a significant effect on digestibility with smaller particles being digested faster due to a higher surface to volume ratio, giving better access by α-amylase [19,20,21]. Semolina particle size is an important parameter of wheat processing and can exert a significant influence on the quality of the semolina and derived finished products. Different particle size ranges can be used depending on customer requirements. However, the role of the semolina particle size distribution used to manufacture the pasta and how that influences the pasta structure and potentially starch digestion is not known to the authors’ knowledge. Both the size of the particles and their composition affected by the milling and sieving process can have variable effects on baking quality and noodle texture [22,23]. In our study semolina was fractionated into a range of particle sizes. Each sieve has its own distribution of particles so that retained on the 250 µm sieve, for example, contains particles between 250–315 µm dictated by the screen aperture limits. As the fraction becomes smaller in average particle size, the water absorption (FWA%) increased from coarse to fine particles in a somewhat linear fashion (Table 1).

Due to the effect of volume to surface area of particles, smaller particles require more water to achieve the same dough consistency. The dough strength measured by DDT and softness at 5 min past peak mixing time was different between the fractions. Compared to the unfractionated control, fractions 180–315 have shorter DDT and fractions < 180–250 have larger softness values, both indicative of weaker dough (Table 1). This is consistent with the superfine grinding of wheat flour where finer particles make a weaker dough with higher water absorption [24]. The protein content of semolina increased moving from coarse to finer particle size consistent with Sacchettia, Cocco, Cocco, Neria, Mastrocola (2011) [25]. Finer fractions contain more endosperm where the majority of the grain protein is located so they would have higher protein contents.

Pasta made from the different semolina particle size fractions shows variation in cooking quality and texture (Table 2).

There was little variation in cooking time except <180 had the shortest which might be related to the more rapid hydration of the spaghetti strands due to pasta formed from finer particles. The <180 pasta had the lowest pasta water uptake (WABS%) possibly due to having less starch (highest protein) to swell compared to the control, consistent with Sacchettia et al. (2011) [25]. Additionally, the <180 pasta had a shorter cooking time because longer cooking leads to more water uptake. This sample had equivalent cooked firmness to the control but coarser fractions like 250, 315 and 425 were less firm than the control and the <180 samples. The highest firmness was obtained with the 180 fraction. There was a tendency for pasta made from finer particles to become firmer, and this was more evident in the F-Area data. This could simply be due to the increase in protein content in the smaller particle size fractions as cooked pasta firmness is highly correlated to protein content [18,26]. While there were significant differences in the overcooking tolerance, these differences are not considered practically different. Cooking loss was lowest in the control sample and only slightly higher in the other samples. Only the <180 pasta had a significantly higher cooking loss, but in practice, it is acceptable. Noodles made from a wider range in flour particle size (17–358 µm) showed increased cooking loss when smaller size particles were used which we did not observe, however, noodles are processed differently to spaghetti and much lower particle size was used by these authors [22]. Pasta WABS% tended to decrease with a smaller particle size fraction which could be related to the higher protein preventing the swelling of the starch. This is opposite to that reported for noodles prepared from wheat flour with particles ranging from 17–358 µm [22]. They suggested that the finer granulation flour has higher starch damage and this would lead to an increase in flour water absorption and thus increased noodle water absorption. While in pasta, despite not having any data on starch damage, we hypothesise that the finer granulation leads to a more compact pasta structure making it more difficult for the pasta to increase in volume compared to larger semolina particles. In addition, pasta made from finer granulation flour had higher protein content, this is likely to provide an extensive and elastic protein network that can prevent or delay starch granules from over-swelling during cooking. Pasta stickiness was minimally impacted with a tendency to decrease although only the 425 pasta was significantly higher. There were no clear trends in dry pasta colour which shows excellent L*, a* and b* values. The 315 pasta had the highest yellowness and brightest pasta while <180 was the least and also with the highest redness compared to the control.

The semolina particle size distribution can vary depending on the milling process used with a tendency over the years for the industry to produce a finer granulation for pasta making although a broader range of particles is still used by many high quality pasta producers. Given that different semolina particle size ranges can be used to make pasta, it was of interest to see if this can alter pasta starch digestion. The starch digestion curves for the pasta with different particle sizes show the typical curve seen in spaghetti with two different digestion rates decreasing over time [10]. The curves show differences between samples and the AUC values indicate significant differences in the extent of starch digestion (Figure 1).

Pasta with 315 was digested to the greatest extent (AUC 3883) than all other samples while pasta with <180 was the least digested (AUC 3462) but not different to 180 and 250 while 180, 250 and 425 pasta had similar AUC to control. One possible explanation of why the <180 was the least digested is that the smaller particle size flour interacts more strongly with the gluten protein during pasta manufacture leading to a more dense structure which would reduce starch digestion. It was noted that there was a trend for AUC to decrease with an increase in pasta protein content except for the 425 pasta indicating this might be another factor affecting AUC besides particle size differences. Although a broader range of protein might be needed to influence the extent of starch digestion in pasta [18] compared to the range studied here. The 315 pasta also had the lowest cooked firmness while the <180 pasta was the firmest, which if this is indicative of the pasta structure, suggests creating a strong, elastic and extensive protein network around the starch granules could have influenced the extent of starch digestion, lowering it. It should be noted that the changes in the AUC while significant between <180 and 315 pasta is small compared to the effect of protein content and selected fibre additions on in vitro starch digestion [18,27]. Further work is needed to better understand the mechanism as to why semolina particle size distribution impacts pasta microstructure and potentially, pasta GI. The data presented suggest that consideration could be given to particle size as a way to manipulate pasta microstructure and starch digestion in light of the importance of carbohydrate food and GI.

### 3.2. Effect of Spaghetti Extrusion Conditions on Starch Digestion and Technological Quality

Three extrusion parameters were evaluated for their impact on pasta quality and starch digestion. The water level was kept constant although it has been shown previously that water absorption and barrel temperature have the greatest effect on the pasta quality [28]. Pasta made using a twin-screw extruder produced larger diameter spaghetti that had a much rougher surface than that made with the single-screw extruder with a smaller diameter in previous studies because a larger non Teflon coated die was used. As a result, cooked firmness values are much higher in the twin-screw extruded pasta as indicated in Table 3. Additionally, the twin-screw extruded pasta has a much lower stickiness and WABS and higher CL than single-screw extruded pasta prepared in this study with a rougher surface consistent with a previous report [29] (compare with pasta results made with a single screw extruder in Table 2).

Varying worm (screw) speed caused significant effects on all pasta quality traits except WABS. As screw speed increased pasta became less firm from 2068 g to 1325 g with no significant difference between 250 and 300 rpm. However, overcooking tolerance improved at a speed above 200 rpm. Higher screw speeds will reduce the time for the gluten polymer to develop and this would impact the starch granule-gluten network making it easier to swell and consequently less firm after cooking. The pasta was stickier using 150 rpm then decreased with no difference between the other speeds. Cooking loss increased with extrusion speed to unacceptable levels above 250 rpm probably because the dough is not fully developed so the starch-gluten matrix is not providing enough encapsulation of the granules so they can be more readily gelatinized and this leads to more solids lost during cooking. Pasta brightness (DPL*) progressively increased with increasing speed, which is desirable but redness (DPa*) also increased affecting the visual appeal while yellowness (DPb*) only increased above 200 rpm. The data indicates that a speed of 200–250 rpm produces acceptable pasta with good firmness that is bright with low stickiness and marginal cooking loss. Higher screw speeds will reduce extrusion time and limit damage to the gluten network caused by high barrel temperatures that could cause proteins to be unable to aggregate.

Varying die temperatures from 60 to 85 °C caused significant effects on all pasta quality traits except SPH and CL (Table 3). Variable effects were seen and for firmness, this was highest using 85 °C and lowest at 75 °C with no clear trend. Interestingly, at 75 °C pasta resilience was greatest (lowest overcooking tolerance) and the least at 85 °C (possibly due to partial starch gelatinization) with no differences between 60 °C and 70 °C. Pasta brightness decreased slightly above 60 °C then did not change significantly while redness and yellowness increased at the higher temperatures. It is known during pasta drying and cooking of foods at high temperatures, non-enzymatic browning, caused by chemical reactions between gluten proteins and reducing sugars, can occur leading to higher redness. This may be happening during the higher temperature extrusion. Stickiness (S-Area) was higher for 70 °C but there were no significant differences in S-PH between samples. Cooking loss tended to increase with temperature but was only significantly higher at 85 °C vs. 60 °C and did not differ between other treatments while WABS decreased to what would be considered a low level only at 85 °C. A much larger increase in pasta cooking loss (250%) was reported with extrusion cylinder temperature increasing from 35 °C to 70 °C [8]. Generally, extrusion temperatures between 40–50 °C are considered optimal for pasta-making as there would not be significant protein denaturation and starch gelatinization and these temperatures would facilitate extrusion by lowering dough viscosity.

Pressure varies during the movement of the dough along the screw being highest at the die. Varying die pressure from 14–32 bar caused significant effects on only firmness and pasta colour (Table 3). Firmness increased as pressure increased to 28 bar with no further rise at 32 bar. Overcooking tolerance changed from 27.8 to 14.6 for die pressures of 14 and 32 bar, respectively. There were no clear trends with pressure on dry pasta colour. These data provide more information about the effect of extrusion variables on pasta quality since this has not been exhaustively investigated.

There were no significant effects of all three extrusion parameter variations studied on the extent of pasta starch digestion measured by AUC (Table 3). This data is supported by Fardet et al. (1999) [30] who found no change in the in vitro starch hydrolysis in extruded cooked pasta between a screw speed of 150 or 250 rpm or die temperature of 40 °C or 70 °C. It could be that the conditions studied might have affected the fresh pasta digestibility that was not examined, as the pasta was dried at a temperature up to 60 °C, so any effect could have been negated due to the structuring of the pasta at high temperature drying and also during the cooking because these processes create a denatured protein and starch matrix [9,12].

### 3.3. Effect of Transglutaminase (TG) Addition in Pasta Making on Technological Quality and Spaghetti Starch Digestion

Transglutaminase can lead to a high level of S-S cross-linking in the gluten network during the manufacture of pasta via catalysis of the isopeptide crosslinks between ε-amino groups of lysine residues and β-carboxyamide groups of glutamine residues present in gluten and can lead to a more intense encapsulation of the starch granules [17,31]. This could potentially decrease the susceptibility of the pasta to α-amylase action and lower the percentage of starch digested in the cooked product during digestion in the human digestive tract. The impact of adding transglutaminase in the pasta preparation was performed at a range of protein levels given the importance of proteins in the reaction. The FWA% and protein content of the commercial semolina sample was 57% and 13% (14% mb), while the samples of low protein were 49.7%, 9.53% and high protein, 58%, 16.88%, respectively. Cooking times of spaghetti samples varied from 12 min 30 s to 14 min with a tendency for cooking time to increase slightly with TG present except in the low protein sample (Table 4).

Cooked firmness was lowest for low protein and highest for the high protein samples, as expected since firmness is highly correlated to the protein content of semolina [18,26]. TG decreased firmness (peak height and area) in the commercial and low protein samples but had the opposite effect in the high protein sample (Table 4) which could be because the higher gluten content allowed more crosslinks in the high protein pasta. Aalami and Leelavathi (2008) [32] found higher pasta firmness from TG in weak gluten pasta which is in contrast to our findings. The low protein semolina used was previously found to have weak dough properties [18]. However, TG did improve overcooking tolerance in the low protein pasta making it comparable to high protein pasta with no significant differences in overcooking tolerance in the other samples. Given TG’s role in the formation of cross-links, this might account for the improved elastic recovery of the pasta. Cooked spaghetti stickiness is related to the proportion of surface material rinsed from the cooked spaghetti after draining [33] with low stickiness desirable. While there were differences in pasta stickiness between samples, the range was narrow, 16.7–23.3 g (S-PH) with the low protein pasta being significantly more sticky. It is known that higher protein content creates a starch-gluten structure that reduces solids loss during cooking resulting in a decrease in surface stickiness [9]. There was no effect of TG on stickiness except in the low protein pasta for S-Area only, increasing stickiness marginally. The amount of material lost in the cooking water is also another measure of pasta quality and this needs to be minimal with <7% generally considered acceptable. A high cooking loss was obtained in low protein pasta with no differences between the commercial and high protein pasta (Table 4). The quantity of protein is an important contributor to forming a starch-gluten matrix and higher protein content will aid with low cooking loss as was found in this study. TG increased cooking loss in the low protein and commercial pasta samples but not in the high protein pasta. Aalami and Leelavathi (2008) [32] reported a reduced cooking loss in spaghetti with TG addition while Sissons et al. (2010) [17] reported no effect using TG but in a semolina of high protein content, consistent with the data in Table 4. The data suggests protein has a larger impact on cooking loss than TG. There were also differences in the water absorption of pasta (weight gain from absorbing water during the cooking) with a low protein having the highest WABS. There was no significant effect of TG on this trait consistent with a previous study [17].

TG had no significant effect on the extent of starch digestion with no differences in the AUC between samples although the data shows a tendency for higher AUC when TG was included in the pasta formulation but this was not significant (Table 4). Other work [34] implicated TG in reducing the predicted glycemic index in noodles. Overall, there is no clear benefit to adding TG at the dose used (0.5% *w*/*w*) to either improve pasta technological properties or reduce starch digestion extent.

### 3.4. Effect of Spaghetti Cooking Time on Starch Digestion

During pasta cooking water progressively moves into the interior of the spaghetti strand where it causes starch gelatinization and protein coagulation which leads to a continuous network entrapping the starch granules. Once cooked, the starch is fully gelatinized [9] so undercooked pasta is expected to have lower starch digestion than optimally cooked pasta as α-amylase catalyses the hydrolysis of fully gelatinised starch in preference to starch granules [35]. However, the pasta once cooked, has three differentiated internal structures described as external, intermediate and central regions [9,12] with a limited extent of gelatinisation within the central region due to limited water penetration. This manifests as visible (by microscopy) starch granules in the central region. Overcooking the pasta should result in more gelatinisation within this region and should lead to a higher starch digestion extent.

The RS content of uncooked pasta (0.35% dmb) after cooking increased from 1.19% in undercooked to 2.49% in 10 min overcooked. Cooking pasta or noodles has been shown to increase RS [36,37] or have no effect [38,39]. Across the protein range of the samples, starch is digested to a greater extent with overcooking compared to undercooking, as expected (Figure 2). Longer cooking leads to more starch gelatinization making the starch more accessible to the α-amylase used in the assay [39]. Only in the 8.6% and 17% protein pasta were there significant differences in the AUC (Figure 2).

Generally, as protein content increased, starch digestion extent (AUC) decreased whether the pasta was over or undercooked. Interestingly, the high protein-overcooked (OC) pasta has a similar AUC to undercooked (UC) pasta at 11.6 and 8.6% showing that high protein can help reduce starch digestion. The lowest protein pasta was the most digested when overcooked and 17% pasta was the least when undercooked (Figure 2). This suggests that higher protein pasta can reduce the higher glycaemic effect of overcooking pasta, something many consumers do inadvertently. Cooking pasta to an Italian al dente which is cooking less time than FCT would allow for a lower GI pasta, especially if the protein content is high. While consumers in Italy appreciate the al dente pasta texture, other cultures, like Asia prefer a softer texture in line with their long-standing experience consuming noodles. Sissons et al. (2021) [18] found that as semolina protein content increased the in vitro starch digestion initial rate increased while the extent of digestion decreased in line with the data here. Perhaps consumers could look for a high protein pasta (>15 + %) and reduce the cooking time to more like traditional Italian al dente texture rather than fully cooked or sometimes overcooked pasta preferred by many non-Italian consumers worldwide as a strategy to reduce GI. Human clinical trials would be needed to confirm this effect.

### 3.5. Effect of Spaghetti Storage Conditions after Cooking on Starch Digestion

Commercially dried pasta is cooked before use to make it palatable but sometimes the pasta is not consumed immediately and is instead stored refrigerated and either eaten cold or subsequently re-heated prior to consumption. This may affect the starch digestibility. It is known that during the cooling and storage of cooked starchy foods, retrogradation of the starch occurs and becomes resistant to hydrolysis [40]. Storage of cooked food under refrigeration may lead to a reduction in its digestibility and in vitro estimated glycaemic index [41]. However, refrigeration of boiled dry-type spaghetti for 24 h did not have a significant effect on in vitro starch digestibility [19]. It was also found in a human study there was no difference in postprandial glycaemic responses to freshly cooked pasta compared with pasta that was cooked, chilled and then reheated [42]. In our study cooked pasta was stored for up to 72 h and RS increased from 1.42% in fully cooked pasta to ~1.6–1.95% after refrigeration, a small increase that might still lead to lowering starch digestion. The data shows in the starch digestion curves the lowest extent of starch digestion (AUC) occurred in the freshly cooked pasta (FCT 0 h) and the highest after 48 h of refrigerated storage, indicating cold storage did not reduce the extent of starch digestion as shown in the AUC values (Figure 3). Studies performed on other carbohydrate foods suggest refrigeration prior to re-heating can reduce starch digestibility [43] but there are relatively few studies to aid understanding of the mechanism of the starch structural changes that occur under re-heating conditions making it difficult to understand the effects shown in Figure 3.

## 4. Conclusions

There are many factors that can affect the technological quality and starch digestion of pasta and this study only considers a few aspects. While large changes in starch digestion and GI can be achieved by substituting semolina with various fibres, resistant starch, gums, etc. processing has less dramatic effects although literature in this field is limited especially the effect of extrusion variables on pasta quality. The studies presented show some potential to modulate the extent of starch digested but further studies are needed to confirm the in vivo effect using clinical evaluation to determine GI. While clear changes in pasta technological properties were obtained by adding TG, varying extrusion conditions or using different semolina particle size distribution to make the pasta, impacts on the in vitro starch digestion extent were either not observed or there were only subtle reductions in the case of finer granulation semolina possibly related to the amount of gluten and a compact structure. The data has suggested that reducing the cooking time to more like traditional Italian al dente texture rather than fully cooked or sometimes overcooked pasta could be a strategy to lower pasta GI, especially in high protein pasta. Further work is needed to elucidate the mechanisms of how semolina particle size influences the starch-gluten matrix under different protein and cooking conditions.

## Figures and Tables

**Figure 1 foods-11-03650-f001:**
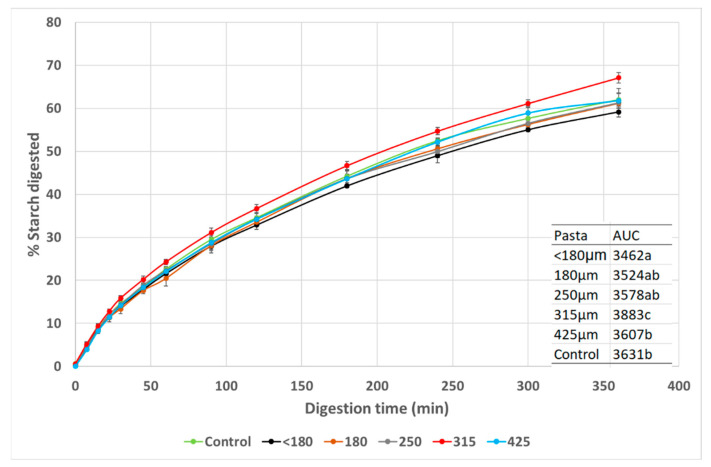
Effect of semolina particle size on pasta starch digestion curves. Data points are mean values (n = 3) with error bars as SD.

**Figure 2 foods-11-03650-f002:**
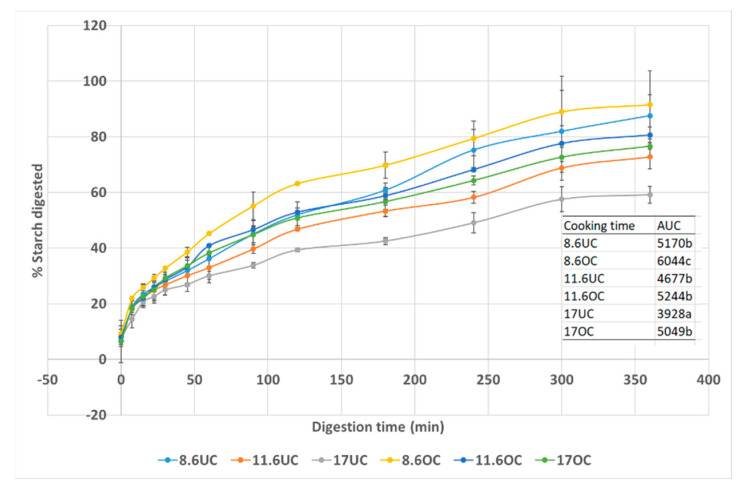
Effect of cooking time in three spaghetti samples with a range in protein content (8.6, 11.6, 17.0%) on starch digestion curves. Data points are mean values (n = 2) with error bars as SD. UC = undercooked pasta FCT-5 min; OC = overcooked pasta FCT + 10 min; AUC=area under the starch digestion curve.

**Figure 3 foods-11-03650-f003:**
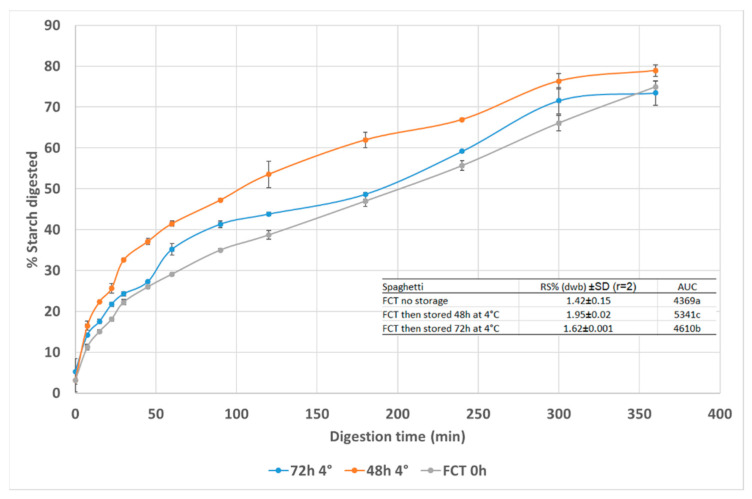
Effect of cold storage duration of fully cooked spaghetti on starch digestion curves. Data points are mean values (n = 2) with error bars as SD. RS% = resistant starch content; AUC = area under the starch digestion curve.

**Table 1 foods-11-03650-t001:** Effect of semolina particle size on semolina and dough properties.

Fraction	Semolina	Farinograph		
	Protein (14% mb)	FWA% (14% mb)	DDT (min)	Softness at 5 min
<180 μm	14.51 f	65.95 f	3.89 de	74.75 f
180 μm	13.93 e	61.55 e	2.69 b	57.15 e
250 μm	13.27 d	57.9 d	2.34 a	34.40 d
315 μm	12.61 b	54.55 b	3.27 c	11.25 b
425 μm	12.32 a	53.25 a	4.18 e	0.25 a
Control	13.11 c	57.1 c	3.93 de	19.65 c
LSD (av.)	0.09	0.38	0.38	7.44
*p*	<0.001	<0.001	<0.001	<0.001

DDT = dough development time. Data are mean of two replicate analyses; Control = unfractionated semolina; Different letters in the same column indicate statistically significant differences at *p* < 0.05.

**Table 2 foods-11-03650-t002:** Pasta made from the different semolina fractions and the effect on pasta technological attributes.

Pasta	FCT (min)	F-PH (g)	F-Area (g/s)	Overcook Tolerance	CL%	WABS%	S-PH (g)	S-Area (g/s)	DPL*	DPa*	DPb*
<180 μm	10.5	867 c	382 c	31.0 bc	6.0 b	123.9 a	20.9 a	10.1 ab	58.80 a	5.24	31.25 a
180 μm	11	926 d	384 c	32.3 c	5.2 a	131.1 b	21.2 a	9.6 a	63.05 bc	3.22	35.29 c
250 μm	12	820 b	339 b	26.3 a	5.4 ab	141.8 c	21.0 a	10.6 ab	65.26 cd	1.84	34.98 b
315 μm	12	780 a	317 a	26.4 a	5.4 ab	147.9 d	22.6 a	12.0 abc	66.46 d	−0.48	39.01 f
425 μm	12.5	792 ab	323 a	27.0 ab	5.5 ab	150.2 d	25.6 b	13.8 c	63.23 bc	−0.41	37.43 e
Control	12	883 c	371 c	29.8 abc	4.9 a	140.4 c	22.5 a	12.2 bc	61.95 b	1.63	36.46 d
LSD		34.43	14.4	4.011	0.6794	2.329	2.83	2.503	2.486	0.4006	0.013
*p*		<0.001	<0.001	<0.05	0.07 NS	<0.001	<0.05	<0.05	<0.001	<0.001	<0.001

Control = unfractionated semolina; NS = not significant; Different letters in the same column indicate statistically significant differences at *p* < 0.05.

**Table 3 foods-11-03650-t003:** Effect of pasta extrusion parameters (screw speed, die temperate and pressure) on pasta technological attributes and the extent of starch digestion. A twin-screw pasta extruder was used.

Variable	Firmness		Overcook Tolerance	Dry Pasta Colour			Stickiness		Cooking Quality		Starch Digestibility
Screw Speed (rpm)	F-PH (g)	F-Area (g/s)		DPL*	DPa*	DPb*	S-PH(g)	S-Area (g/s)	CL(%)	WABS(%)	AUC
150	2068 c	1582 c	ND	58.96 a	0.77 a	10.02 a	14.3 b	6.5 b	6.8 a	114.5 a	4160 a
200	1817 b	1362 b	40.11 b	60.93 b	1.03 b	10.37 a	10.4 a	4.8 a	7.6 ab	120.0 a	4453 a
250	1380 a	974 a	27.26 a	61.79 c	1.38 c	11.30 b	10.4 a	3.7 a	8.5 b	120.1 a	4234 a
300	1325 a	955 a	29.89 a	62.41 d	1.61 c	12.60 c	10.0 a	4.3 a	10.2 c	123.9 a	4212 a
LSD	82.1	74.1	9.105	0.585	0.28	0.79	2.55	1.29	1.02	16.06	1031
*p*	<0.001	<0.001	<0.05	<0.05	<0.05	<0.05	<0.05	<0.05	<0.05	0.45 NS	0.81 NS
Die Temp. (°C)											
60	1652 b	1275 b	33.7 b	60.46 b	1.31 b	11.64 a	11.7 ab	4.9 a	6.8 a	130.5 b	4907 a
70	1660 b	1181 b	37.8 bc	59.39 a	1.12 a	12.95 b	14.2 b	8.0 b	9.2 ab	133.8 b	5078 a
75	1356 a	778 a	11.1 a	58.54 a	1.38 c	13.71 c	10.8 ab	5.2 a	10.1 ab	130.9 b	5601 a
85	2282 c	1476 c	42.2 c	58.61 a	1.44 c	16.05 d	9.9 a	4.9 a	11.9 b	111.6 a	4596 a
LSD	91.7	138.7	5.754	0.91	0.07	0.29	3.76	1.91	4.97	11.52	1956
*p*	<0.001	<0.001	<0.001	<0.05	0.001	<0.001	0.12 NS	<0.05	0.16 NS	<0.05	0.52 NS
Die Pressure (bar)											
14	1151 a	892 a	27.8 ab	58.78 a	2.09 b	13.02 b	9.3 a	3.2 a	7.0 a	131.5 a	4858 a
22	1293 b	947 a	30.6 b	60.79 c	1.54 a	11.33 a	9.5 a	3.0 a	8.0 a	124.2 a	5089 a
28	1587 c	1220 b	28 ab	60.76 c	12 c	11.11 a	9.7 a	3.1 a	7.5 a	123.1 a	5224 a
32	1561 c	1228 b	14.6 a	59.81 b	1.57 a	12.62 b	10.0 a	3.7 a	7.2 a	121 a	5304 a
LSD	112.2	68.8	15.2	0.27	0.19	0.85	1.30	1.03	3.01	21.64	1713
*p*	<0.001	<0.001	0.14 NS	<0.001	<0.05	<0.05	0.68 NS	0.40 NS	0.71 NS	0.53 NS	0.84 NS

ND = not determined due to insufficient sample; NS=not significant; F-PH = cooked pasta firmness peak height; F-Area = cooked pasta firmness area under the curve; DPL* = dried pasta brightness; DPa* = dried pasta redness; DPb* = dried pasta yellowness; S-PH = cooked pasta stickiness peak height; S-Area = cooked pasta stickiness area under the curve; CL = cooking loss; WABS = pasta water absorption; AUC = area under the starch digestion curve; Different letters in the same column indicate statistically significant differences at *p* < 0.05.

**Table 4 foods-11-03650-t004:** Effect of including transglutaminase (TG) at 0.5% (*w*/*w*) in the pasta formulation at three different protein levels on pasta technological attributes and the extent of starch digestion.

Pasta	FCT (min:s)	F-PH (g)	F-Area (g/s)	Overcooking Tolerance	S-PH (g)	S-Area (g/s)	Cooking Loss (%)	WABS	AUC
Commercial semolina	12:30	795 c	350 e	25.9 ab	19.8 abc	9.0 ab	5.6 a	144 a	4464 a
Commercial + TG	13:00	774 c	341 d	28.6 bc	16.7 a	8.0 a	6.5 b	148 ab	4810 a
Low protein	12:30	638 b	259 b	31.7 c	23.1 cd	11.9 b	7.2 c	161 c	4612 a
Low protein + TG	12:30	536 a	211 a	24.3 a	23.3 d	15.1 c	9.4 d	162 c	4700 a
High protein	13:00	823 d	330 c	24.6 a	19.3 ab	9.5 ab	5.7 a	152 b	4121 a
High protein + TG	14:00	931 e	379 e	26.6 ab	22.3 bcd	8.6 a	5.3 a	152 b	4200 a
Sample LSD		21.40	10.53	3.29	3.46	3.16	0.47	4.4920	1006
*p*		<0.001	<0.001	<0.05	<0.05	<0.05	<0.001	<0.001	NS

FCT = fully cooked time; F-PH = cooked pasta firmness peak height; F-Area = cooked pasta firmness area under the curve; S-PH = cooked pasta stickiness peak height; S-Area = cooked pasta stickiness area under the curve; WABS = pasta water absorption; AUC = area under the starch digestion curve; NS=not significant; Different letters in the same column indicate statistically significant differences at *p* < 0.05.

## Data Availability

Data is contained within the article or Appendix A.

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
