# Peer review of "Influence of Some Spaghetti Processing Variables on Technological Attributes and the In Vitro Digestion of Starch"

_foods, 2022, doi:10.3390/foods11223650_

Round 1
Reviewer 1 Report
The paper investigates several process factors such as adding transglutaminase, cooking time, cold storage, extrusion and particle size of flour, to see if they can impact the starch digestion in vitro in spaghetti. But the results show that only particle size has effects on starch digestion, such result is obvious and does not require experimental verification. The research significance of the article is not prominent. Some contents should be modified as this:
1. In abstract, the first sentence lacks a full stop and there are grammatical problems in the second and fourth sentences. Useful results should be mentioned first.
2. The logic of the author's analysis of the experimental results needs to be improved.
There are no useful conclusion for readers in the paper, all factors except partical size have no effects of starch digestion, the effects of finer semolina particle size on starch digestion is obvious, No experimental verification is required. This is all reason.
Author Response
We would like to thank the reviewer for their suggested improvements. Please see our responses to each point raised and what we have changed.
In regards to reviewer comment "But the results show that only particle size has effects on starch digestion, such result is obvious and does not require experimental verification. The research significance of the article is not prominent."
We have noted in the text section 3.1 “size and shape of the pasta subject to starch digestion has a significant effect on digestibility with smaller particles being digested faster due to a higher surface to volume ratio, giving better access by α-amylase (3 references provided)” So while the reviewer is correct this has only been evaluated by digesting different size pasta pieces, either long, short or crushed. The role of the semolina particle size distribution used to make the pasta and this effect on the starch digestion is not known to the authors knowledge. It was stated by Petitot et al [9] that “a modification in the pasta structure through a pasta processing change could have an impact on the rate of starch degradation, i.e. the pasta GI.” This reference did not cover the topic of how varying semolina particle size range affects pasta structure and properties. The semolina particle size distribution can vary depending on the milling process used with a tendency over the years for industry to produce a finer granulation for pasta making although a broader range in particles is still used by many high quality pasta producers. Given that different semolina particle size ranges can be used to make pasta, it was of interest to the authors to see if this can alter pasta starch digestion by changing the pasta structure. Given the amount of processing involved in making pasta and its cooking, it is not surprising that the effect on starch digestion we observed, although not large, did occur and we feel this is important information for the pasta industry in light of the importance of carbohydrate food and GI. The comment from the reviewer “particle size has effects on starch digestion, such result is obvious and does not require experimental verification” does not easily explain results in Fig 3 (now Fig 1 in revision). Indeed, the finest granulation, <180 fraction was the least digested and the coarsest was not the fastest digested. So it is not so obvious what is happening and deserves the experimentation. It could be said that a smaller particle size semolina will allow a more dense pasta structure with a stronger bond to the protein during its manufacture and this would lead to a reduction in starch digestion. However, this does not explain why the coarser particle size pasta was not digested to the greatest extent, the 315 pasta was. Clearly, the argument on particle size is not so straight forward.
We have modified the text to “One possible explanation of why the <180 was the least digested is that the smaller particle size flour interacts more strongly with the gluten protein during pasta manufacture leading to a more dense structure which would reduce starch digestion.” “The semolina particle size distribution can vary depending on the milling process used with a tendency over the years for industry to produce a finer granulation for pasta making although a broader range in particles is still used by many high quality pasta producers. Given that different semolina particle size ranges can be used to make pasta, it was of interest to the authors to see if this can alter pasta starch digestion.”
- In abstract, the first sentence lacks a full stop and there are grammatical problems in the second and fourth sentences. Useful results should be mentioned first.
We have revised the abstract and moved the positive results forward
- The logic of the author's analysis of the experimental results needs to be improved.
We have re-ordered the R&D into semolina particle size-extrusion-TG-cooking time-storage to be more logical and align with the pasta process.
There are no useful conclusion for readers in the paper, all factors except partical size have no effects of starch digestion, the effects of finer semolina particle size on starch digestion is obvious, No experimental verification is required. This is all reason.
The other positive effect on starch digestion we provide is cooking time. The implication is overcooking can negatively impact starch digestion and potentially the GI of the cooked pasta, but that would need proof using human subjects. But it does raise the point that the benefit of the low-moderate pasta GI could be reduced if consumers overcook pasta and they probably are not aware of this. Hence it is a useful result worth reporting. Even if a result is negative, its still useful information. Indeed, (Bresciani, A.; Pagani, M.A.; Marti, A. Pasta-Making Process: A Narrative Review on the Relation between Process Variables and Pasta Quality. Foods 2022, 11, 256. https://doi.org/10.3390/ foods110302561) noted “On the other hand, the effect of processing variables (i.e., hydration level, extrusion pressure/temperature/mechanical energy) on pasta quality has not yet been exhaustively investigated.” Shows that the results we provide in Table 3 are worth reporting adding to the limited literature on this topic alone in addition to effects on starch digestion. Also, there are no reported studies investigating the impact of transglutaminase present in the pasta manufacture on the in vitro starch digestion and so this data we presented is useful to have even if we found no effect. This was mentioned.
We have modified the text “processing has less dramatic effects although literature in this field is limited especially the effect of extrusion variables on pasta quality.”
Reviewer 2 Report
I have been asked to review the manuscript entitled “Influence of some Spaghetti Processing Variables on Technological quality and in vitro Digestion of Starch” with a Manuscript ID of foods-2009154.
In this study, the authors investigated the effect of various variables in the production and cooking of spaghetti such as semolina particle size, transglutaminase addition, extrusion conditions, cooking and storage conditions on in vitro starch digestion.
Here are my commnens;
Line 103: Is Deltagen Australia, Melbourne, Australia the supplier of Megazyme kits? Megayme Inc. is the supplier I guess.
There are several mistakes regarding putting the space between the number and its unit. Some (but not all of them) are listed below, the author’s should check the whole manuscript accordingly;
Line 91: 50g
Line 92: 650FU
Line 116: 13h
Line 117: 4h
Line 146: 100 °C
Line 167: 0.02M
Line 173: 5ml
Line 239: 40g/min
Line 253: 180μm
There are some mistakes in the Tables
Table 1, 2, 3, 4: the letters near the numbers indicate the statistical differences, however, an explanation needs to be put under the tables.
Table 1; p values are written in italic in the other tables, however, in Table 1 they are not.
Table 1; Cooking loss%; % should be in paranthesis
Author Response
We would like to thank the reviewer for their suggested improvements. Please see our responses to each point raised and what we have changed.
Line 103: Is Deltagen Australia, Melbourne, Australia the supplier of Megazyme kits? Megayme Inc. is the supplier I guess.
Yes in Australia Deltagen supplies the Megazyme kits. No changes
There are several mistakes regarding putting the space between the number and its unit. Some (but not all of them) are listed below, the author’s should check the whole manuscript accordingly;
Line 91: 50g
Line 92: 650FU
Line 116: 13h
Line 117: 4h
Line 146: 100 °C
Line 167: 0.02M
Line 173: 5ml
Line 239: 40g/min
Line 253: 180μm
All these changes have been made and the entire document has been checked
There are some mistakes in the Tables
Table 1, 2, 3, 4: the letters near the numbers indicate the statistical differences, however, an explanation needs to be put under the tables.
This statement has been added “; Different letters in the same column indicate statistically significant differences at p < 0.05.”
Table 1; p values are written in italic in the other tables, however, in Table 1 they are not.
Only Table 2 has italics for P and LSD. We made this normal text to align with the other tables
Table 1; Cooking loss%; % should be in paranthesis
Corrected
Reviewer 3 Report
Comments to Author:
The manuscript needs some improvements concerning the language and updated literature. Following are the main suggestions:
Abstract
1. “Addition of Transglutaminase in the pasta formulae had minimal impacts on quality and no effect on starch digestion over a wide protein range.” is meaningless.
2. “Cooking time has...” should be “Cooking time had...”.
3. “Varying three extrusion parameters affected technological quality but none had any impact on the extent of starch digested.” should be reorganized.
4. “The semolina particle size distribution impacted pasta quality and starch digestion to a small extent indicating a finer semolina particle size may promote a more compact structure and help to reduce starch digestion.” should be reorganized.
5. Lack of relevant data support in Abstract.
Introduction
6. Line 40. “GI” should not be abbreviated for the first time.
7. Lines 48 and 49, “630 µm to <125 µm or narrower range”?
8. “The purpose of this study was to evaluate the impact of various variables in the production and cooking of spaghetti (semolina particle size, transglutaminase addition, extrusion conditions, cooking and storage conditions) on in vitro starch digestion to see if optimisation of low starch digestibility can be further achieved with any of these modifications.” should be reorganized.
9. What is “the technological quality”?
Materials and Methods
10. “Microbial transglutaminase (glutaminyl-peptide-amine γ-glutamyltransferase, E.C. 2.3.2.13) a gift from Swift, of IMCD Australia (Veron TG, Mulgrave, Australia).” should be reorganized.
11. “The freeze-dried gluten was ground using a coffee grinder and sieved (500 μm mesh) and used to increase the protein content of semolina as required.” should be reorganized.
12. Line 90, what is “as-is”?
13. “In brief, mixing of semolina and water begins in a premixing chamber for 15 min then the crumbly dough is transferred to the vacuum and temperature controlled (45°C) extruder fitted with a 1.82 mm teflon coated spaghetti die (Appar Laboratorio, Rome).” should be reorganized.
14. Line 115-118, “rh” should be “RH”?
15. Line 131, “which had...” ?
16. “A force of 500 g is used with a 5 kg load cell.” should be reorganized.
17. Line 172, “210U”?
18. Line 213, “RS” should not be abbreviated for the first time.
Results and Discussion
19. Line 272, “(19, 17)” should be “(17, 19)”.
20. “The low protein semolina used was found to have weak dough properties previously.” should be reorganized.
21. “This suggests TG improved the elastic recovery of pasta which could be related to the formation of more cross-links.” should be reorganized.
22. “However, Li, Tan, Liong, Easa, (2014) (23) found that noodles prepared from wheat flour plus soy protein isolate and TG (0.5% w/w) reduced the predicted glycemic index compared to wheat flour only noodles but it is unclear whether the soy or TG was responsible or there was an interaction leading to the reduction although authors claim the lower pGI was due to the TG with no direct evidence.” should be reorganized.
23. “It was however noted that in low protein semolina, pasta overcooking tolerance was improved but cooking loss was unacceptably high due to the low protein content.” should be reorganized.
24. “Some studies with pasta or noodles have shown that heating increases RS (25, 26) while others find no change in RS content (27, 28).” should be reorganized.
25. Lines 355 and 156, OC and UC should not be abbreviated for the first time.
26. “The data shows in the starch digestion curves the lowest extent of starch digestion (AUC) occurred in the non-stored pasta (FCT 0h) and the highest after 48 h and less so after 72 h refrigerated storage, indicating cold storage did not reduce the extent of starch digestion as shown in the AUC values (Fig. 2).” should be reorganized.
27. Line 413, “2068g to 1325g” should be “ 2068 g to 1325 g”.
28. “The data indicates that a speed of 200-250 rpm produces acceptable pasta with good firmness that is bright with low stickiness and marginal cooking loss. Higher screw speeds will reduce extrusion time and limit damage to the gluten network caused by high barrel temperatures that could cause proteins being unable to aggregate.” should be reorganized.
29. “In our study semolina was fractionated into a range of particle sizes Each sieve has its own distribution of particles so that retained on the 250 µm sieve for example, contains particles between 250-315 µm dictated by the screen aperture limits.” should be reorganized.
Conclusions
30. “The studies presented show some potential to modulate the extent of starch digested but would need confirmation using clinical evaluation of product GI.” should be reorganized.
31. “While clear changes in pasta technological properties were obtained by adding TG, varying extrusion conditions or using different particle size distribution of the semolina to make the pasta, impacts on the in vitro starch digestion extent were either not obtained or only subtle reductions in the case of finer granulation semolina possibly related to the amount of gluten and a compact structure.” should be reorganized.
Author Response
Comments to Author:
The manuscript needs some improvements concerning the language and updated literature. Following are the main suggestions before publication:
We would like to thank the reviewer for their suggestion for improvement and our responses are shown below each query with the modification to the manuscript text within “xxxx”.
Abstract
- “Addition of Transglutaminase in the pasta formulae had minimal impacts on quality and no effect on starch digestion over a wide protein range.” is meaningless.
Response: We are uncertain what the reviewer is asking. There was some effect on pasta quality and that data is worth showing and the reason for including this experiment is explained in section 3.3. We have revised the text to “Transglutaminase in the pasta formulae improved overcooking tolerance in low protein pasta comparable to high protein pasta with no other significant effects and had no effect on starch digestion over a wide protein range”
- “Cooking time has...” should be “Cooking time had...”.
Response: This was deleted in the revised version
- “Varying three extrusion parameters affected technological quality but none had any impact on the extent of starch digested.” should be reorganized.
Response: This has been changed to “Varying three extrusion parameters (die temperature, die pressure, extrusion speed) impacted pasta technological quality but not the extent of starch digestion.”
- “The semolina particle size distribution impacted pasta quality and starch digestion to a small extent indicating a finer semolina particle size may promote a more compact structure and help to reduce starch digestion.” should be reorganized.
Response: This was deleted in the revised version
- Lack of relevant data support in Abstract.
Response: Additional data has been included
Introduction
- Line 40. “GI” should not be abbreviated for the first time.
Response: Abbreviation added “and is known for its low-moderate glycaemic index (GI) (2).”
- Lines 48 and 49, “630 µm to <125 µm or narrower range”?
Response: Revised to “particle size range typically 630 µm to <125 µm (7).”
- “The purpose of this study was to evaluate the impact of various variables in the production and cooking of spaghetti (semolina particle size, transglutaminase addition, extrusion conditions, cooking and storage conditions) on in vitro starch digestion to see if optimisation of low starch digestibility can be further achieved with any of these modifications.” should be reorganized.
Response: Revised to “The purpose of this study was (1) to evaluate the impact of semolina particle size, extrusion conditions and transglutaminase addition on pasta technological properties (texture, cooking quality) and the pasta in vitro starch digestion and (2) the impact of cooking time and storage after cooking on pasta starch digestion to see if starch digestibility can be reduced.”
- What is “the technological quality”?
Response: This is covered in point 8
Materials and Methods
- “Microbial transglutaminase (glutaminyl-peptide-amine γ-glutamyltransferase, E.C. 2.3.2.13) a gift from Swift, of IMCD Australia (Veron TG, Mulgrave, Australia).” should be reorganized.
Response: Revised to “Microbial transglutaminase (glutaminyl-peptide-amine γ-glutamyltransferase, E.C. 2.3.2.13) was a gift IMCD Australia (Veron TG, Mulgrave, Australia).”
- “The freeze-dried gluten was ground using a coffee grinder and sieved (500 μm mesh) and used to increase the protein content of semolina as required.” should be reorganized.
Response: Revised to “Gluten was isolated from a commercial semolina sample, freeze-dried and ground using a coffee grinder and sieved (500 μm mesh) as described previously (13). Gluten was used to increase protein content of semolina to desired levels.”
- Line 90, what is “as-is”?
Response: Revised to “Flour water absorption was determined using a MicroDoughLAB (Perten Instruments, Australia) fitted with a 50 g bowl and mixing at 120 rpm to target peak 650 FU, in duplicate expressed on a 14%mb.”
- “In brief, mixing of semolina and water begins in a premixing chamber for 15 min then the crumbly dough is transferred to the vacuum and temperature controlled (45°C) extruder fitted with a 1.82 mm teflon coated spaghetti die (Appar Laboratorio, Rome).” should be reorganized.
Response: Revised to In brief, semolina was mixed with water in a premixing chamber for 15 min then the crumbly dough transferred to the vacuum and temperature controlled (45°C) extruder, fitted with a 1.82 mm teflon coated spaghetti die (Appar Laboratorio, Rome).”
- Line 115-118, “rh” should be “RH”?
Response: Corrected
- Line 131, “which had...” ?
Response: Deleted
- “A force of 500 g is used with a 5 kg load cell.” should be reorganized.
Response: Changed to “A 5 kg load cell using a force of 500 g was used”
- Line 172, “210U”?
Response: Changed to “(prepared fresh, 135 Units α-amylase, Sigma A6255 from porcine pancereas and 210 Units amyloglucosidase”
- Line 213, “RS” should not be abbreviated for the first time.
Response: This has been defined earlier in section 2.2 as “Total starch and resistant starch (RS) content of ground spaghetti (coffee grinder, sieved across a 250 µm screen) were assayed in triplicate using Megazyme Total Starch and Resistant Starch kits; analysis procedure (a) was used for total starch (Deltagen Australia, Melbourne, Australia).”
Results and Discussion
- Line 272, “(19, 17)” should be “(17, 19)”.
Response: Not relevant as doc had been revised
- “The low protein semolina used was found to have weak dough properties previously.” should be reorganized.
Response: Revised to “The low protein semolina used was previously found to have weak dough properties”
- “This suggests TG improved the elastic recovery of pasta which could be related to the formation of more cross-links.” should be reorganized.
Response: Replaced with “Given TG role in the formation of cross-links, this might account for the improved elastic recovery of the pasta.”
- “However, Li, Tan, Liong, Easa, (2014) (23) found that noodles prepared from wheat flour plus soy protein isolate and TG (0.5% w/w) reduced the predicted glycemic index compared to wheat flour only noodles but it is unclear whether the soy or TG was responsible or there was an interaction leading to the reduction although authors claim the lower pGI was due to the TG with no direct evidence.” should be reorganized. Response: Replaced with “Other work (34) implicated TG in reducing the predicted glycemic index in noodles.”
- “It was however noted that in low protein semolina, pasta overcooking tolerance was improved but cooking loss was unacceptably high due to the low protein content.” should be reorganized.
Response: Deleted
- “Some studies with pasta or noodles have shown that heating increases RS (25, 26) while others find no change in RS content (27, 28).” should be reorganized.
Response: Revised to “Cooking of pasta or noodles has been shown to increases RS (36, 37) or have no effect (38, 39).”
- Lines 355 and 156, OC and UC should not be abbreviated for the first time.
Response: Revised to “Interestingly, the high proteinovercooked (OC) pasta has similar AUC to undercooked (UC)”
- “The data shows in the starch digestion curves the lowest extent of starch digestion (AUC) occurred in the non-stored pasta (FCT 0h) and the highest after 48 h and less so after 72 h refrigerated storage, indicating cold storage did not reduce the extent of starch digestion as shown in the AUC values (Fig. 2).” should be reorganized.
Response: Revised to “occurred in the freshly cooked pasta (FCT 0 h) and the highest after 48 h of refrigerated”
- Line 413, “2068g to 1325g” should be “ 2068 g to 1325 g”.
Response: Deleted previously
- “The data indicates that a speed of 200-250 rpm produces acceptable pasta with good firmness that is bright with low stickiness and marginal cooking loss. Higher screw speeds will reduce extrusion time and limit damage to the gluten network caused by high barrel temperatures that could cause proteins being unable to aggregate.” should be reorganized.
Response: Deleted previously
- “In our study semolina was fractionated into a range of particle sizes Each sieve has its own distribution of particles so that retained on the 250 µm sieve for example, contains particles between 250-315 µm dictated by the screen aperture limits.” should be reorganized.
Response: Deleted previously
Conclusions
- “The studies presented show some potential to modulate the extent of starch digested but would need confirmation using clinical evaluation of product GI.” should be reorganized.
Response: Revised to “The studies presented show some potential to modulate the extent of starch digested but further studies are needed to confirm the in vivo effect using clinical evaluation to determine GI.”
- “While clear changes in pasta technological properties were obtained by adding TG, varying extrusion conditions or using different particle size distribution of the semolina to make the pasta, impacts on the in vitro starch digestion extent were either not obtained or only subtle reductions in the case of finer granulation semolina possibly related to the amount of gluten and a compact structure.” should be reorganized. Response: Deleted previously
Reviewer 4 Report
This manuscript described the impact of various variables in the production and cooking of spaghetti on in vitro starch digestion. The research design of this work is interesting. The author has done a lot of basic work. However, many sentences in the article are too cumbersome to understand, the tense of the article is quite confused, and some observations to the manuscript should be resolved.
The detail comments are as followed:
1. Page 1 Line 13-15, this sentence is hard to understand. And there are several confused sentences in the Abstract.
2. Page 1 Line 20, “has”, keep the tense unified.
3. There are too few latest references are used in the part of Results and discussion, which could not analyze the experimental data and phenomena accurately and reliably.
4. The reason of transglutaminase addition should be emphasized in the introduction.
5. Page 1 Line 253-254, a space is needed after the number.
6. In Tables and Figures, the “a, b, c” should be defined. And the line “P” in Table, what is meaning of the number “.001” in the presence of “0.01” “0.002”, this is very confusing.
7. In Figures, the basic figure quality and content need to be guaranteed.
8. It is best not to cite references in the Conclusions.
9. In the Conclusions lines 67-69, the sentence is same with Part 3.2 Lines 365-367. And the 15+% is confusing in this part.
10. The formation of References is not unified, such as the use or not of “doi”, the abbreviation or full name of the journal.
11. The author introduces these processing techniques, what about the combination of the processing on in vitro starch digestion.
Author Response
We would like to thank the reviewer for their suggested improvements. Please see our responses to each point raised and what we have changed. We have attempted to improve readability throughout the document.
The detail comments are as followed:
- Page 1 Line 13-15, this sentence is hard to understand. And there are several confused sentences in the Abstract.
We have revised this statement and others in the abstract to improve comprehension.
- Page 1 Line 20, “has”, keep the tense unified.
changed
- There are too few latest references are used in the part of Results and discussion, which could not analyze the experimental data and phenomena accurately and reliably.
Of the references cited in results and discussion from #8-43, seven of them were published from 2017 onwards (#18,19,31,32,33,35,39), we consider these to be recent references and sufficient.
- The reason of transglutaminase addition should be emphasized in the introduction.
We have included a statement “Modification of the pasta processing and cooking/storage conditions could alter the pasta structure and consequently influence starch digestibility and technological properties (9) as well as processing aids like transglutaminse (4) and these may be a way to further modulate pasta GI.”
- Page 1 Line 253-254, a space is needed after the number.
corrected
- In Tables and Figures, the “a, b, c” should be defined.
As for reviewer #2 this has been corrected
And the line “P” in Table, what is meaning of the number “.001” in the presence of “0.01” “0.002”, this is very confusing.
Numbers like “.01” have been replaced with <0.05
- In Figures, the basic figure quality and content need to be guaranteed.
Could the reviewer explain what is meant by “guaranteed”
- It is best not to cite references in the Conclusions.
Citations have been removed
- In the Conclusions lines 67-69, the sentence is same with Part 3.2 Lines 365-367. And the 15+% is confusing in this part.
We have revised to “high protein pasta (> 15+%)” We have also changed conclusion to “The data has suggested that reducing the cooking time to more like traditional Italian al dente texture rather than fully cooked or sometimes overcooked pasta could be a strategy to lower pasta GI.”
- The formation of References is not unified, such as the use or not of “doi”, the abbreviation or full name of the journal.
Since we do not have the doi for all references, we have removed this. We decided to provide the full name for the journal and have edited the reference list accordingly
- The author introduces these processing techniques, what about the combination of the processing on in vitrostarch digestion.
Combination effect of processing variables on in vitro digestion is important. However, in this research the experimental plan has not been designed for determination of the combination effects between variables. We intend to look into the presence of any interactions between these variables in the future.
Round 2
Reviewer 1 Report
Authors have modified the paper carefully and I recommand the acceptance of the paper at present status.
Author Response
Thank you for your comments.
Reviewer 3 Report
I agree with the author's reply.
Reviewer 4 Report
The authors did a good job of revising as required.
Author Response
Thank you for your comments.